# Electroretinographic Assessments of Macular Function after Brilliant Blue G Staining for Inner Limiting Membrane Peeling

**DOI:** 10.3390/jcm11216404

**Published:** 2022-10-29

**Authors:** Gaku Terauchi, Kei Shinoda, Kazuma Yagura, Makoto Kawashima, Soiti Celso Matsumoto, Atsushi Mizota, Yozo Miyake

**Affiliations:** 1School of Medicine, Teikyo University, Tokyo 173-8606, Japan; 2Faculty of Medicine, Saitama Medical University, Saitama 350-0495, Japan; 3Matsumoto Eye Clinic, Tokushima 771-1705, Japan; 4Kobe Eye Center Next Vision, Kobe 650-0047, Japan

**Keywords:** brilliant blue G, electroretinogram, epiretinal membrane, inner limiting membrane, macular hole, vitrectomy

## Abstract

Purpose: The purpose of this study was to determine the effect of brilliant blue G (BBG) staining of the inner limiting membrane (ILM) on macular function. Method: Fourteen eyes of 14 patients consisting of 9 men and 5 women who underwent vitreous surgery with ILM peeling were studied. The mean age of the patients was 68.8 ± 9.14 years. Three eyes had a macular hole and eleven eyes had an epiretinal membrane. The ILM was made more visible by spraying 0.25% BBG into the vitreous cavity. The macular function was assessed by recording intraoperative focal macular electroretinograms (iFMERGs) before and after the intravitreal spraying of the BBG dye. The iFMERGs were recorded three times after core vitrectomy. The first recording was performed before the BBG injection (Phase 1, baseline), the second recording was performed after the spraying of the BBG and washing out the excess BBG (Phase 2), and the third recording was performed after the ILM peeling (Phase 3). All recordings were performed after 5 min of light-adaptation and stabilization of the intraocular conditions. The iFMERGs were recorded twice at each phase. The implicit times and amplitudes of the a- and b-wave, the PhNR, and the d-wave were measured. Wilcoxon signed-rank test were used to determine the significance of differences of the findings at Phase 2 vs. Phase 1 and Phase 3 vs. Phase 1. A *p* value < 0.05 was taken to be statistically significant. Results: The average implicit times of the a-wave, b-wave, PhNR, and d-wave were not significantly different in Phase 1, 2, and 3. The average a-wave, b-wave, PhNR, and d-wave amplitudes at Phase 1 did not differ significantly from that at Phase 2 and at Phase 3. Conclusions: The results indicated that the intravitreal injection of BBG does not alter the physiology of the macula, and we conclude that BBG is safe. We also conclude that iFMERGs can be used to monitor the macular function safely during intraocular surgery.

## 1. Introduction

The first report of intraoperative electroretinogram (ERG) recording was reported by Miyake and Horiguchi in 1991 [1,2,3]. They recorded full-field flicker ERGs during strabismus surgery, scleral buckling surgery, and vitreous surgery. They were able to assess the changes in retinal function during the course of the surgeries. On the other hand, the focal macular ERG (FMERG) allowed the assessment of the function of the macula on a layer-by-layer basis. The findings have advanced our understanding of various macular diseases [4]. In 2015, Matsumoto et al. developed a technique for recording intraoperative FMERGs (iFMERGs) [5], and they reported a change in the macular function after core vitrectomy [6] and after scleral indentation during surgery [7].

During intraocular surgery, a clear view of the vitreous and the inner limiting membrane (ILM) of the retina is critical for successful outcomes. Since the report of anterior capsule staining by Horiguchi et al. [8], several staining agents have been developed, e.g., triamcinolone acetonide (TA), to make the vitreous gel and vitreous cortex more visible [9], and trypan blue, indocyanine green (ICG), and brilliant blue G (BBG) to make the ILM more visible. The increased visibility of the ILM made it easier and safer to peel [10,11,12]. However, there are reports on the adverse effects of vital dyes. For example, it has been reported that TA can cause postoperative sterile endophthalmitis [13,14,15,16], and ICG was shown to be toxic for the retina [17,18].

Several electrophysiological assessments of the macular function after surgery with dye-assisted ILM peeling have been reported [19,20,21,22,23,24]. The ERG recordings were performed as early as one month after the surgery, and some investigators reported a reduction in macular function, although it might have been transient. However, no information has been published on the electrophysiological assessment of macular function immediately after surgery. Furthermore, the in situ effect of the dye on the physiology of the retina has not been reported.

Thus, the purpose of this study was to determine the effect of the BBG staining of the ILM on macular function. To accomplish this, we recorded the iFMERGs before, immediately after staining the ILM with BBG, and after the peeling of the ILM at the completion of the surgery.

## 2. Materials and Methods

Patients: Fourteen eyes of 14 patients who underwent par plana vitrectomy (PPV) with ILM peeling were studied. There were 9 men and 5 women whose mean age was 68.8 ± 9.14 years. Three eyes had a macular hole (MH) and eleven eyes had an epiretinal membrane (ERM). The surgeries performed included PPV on 1 eye and PPV combined with cataract surgery on 13 eyes. The ILM was made more visible by injecting 0.3 mL of a 4-fold-diluted 0.25% BBG.

The procedures used conformed to the tenets of the Declaration of Helsinki. The study was an interventional case series, and the procedures were approved by the Ethics Committee of the Teikyo University School of Medicine (Study ID Number: 10-008-2). A signed informed consent was obtained from all participants before the surgery.

Methods: A vitrectomy surgical system with an intraocular pressure (IOP) control and an intraocular illumination system (Constellation Vision System, Alcon Surgical, Fort Worth, TX, USA) was used to perform the surgery. The operating microscope (Model M844, Leica Microsystems, Weltzer, Germany) had a wide-angle observation system (BIOM, Oculus, Weltzer, Germany) to observe the fundus during the surgery and the recordings of the iFMERGs. The iFMERG recordings were performed with the same methods and conditions reported by Matsumoto et al. in 2015 [5]. Briefly, high-flux, light-emitting diodes (LEDs, OSW4XME3C1E, Optosupply, Taiwan) were used for the light stimuli, and the stimuli were delivered to the macula area by a 25 G directional glass optic fiber cable (25 G Directional Laser Probe Synergetics, Bausch Lomb, St. Louis, MO, USA). The size of the stimulus spot was approximately twice the size of optic nerve head, and the stimuli intensity was 160 cd/m^2^. The stimulus was a 4 Hz rectangular stimulus (100 ms light-on and 150 ms light-off). The background illumination was 3 cd/m^2^. A sterilized gold foil monopolar contact lens (Mayo Corporation, Nagoya, Japan) was used to pick up the retinal responses. The reference silver plate electrode was placed on the forehead, and the ground electrode was placed on the ear lobe. The iFMERGs were amplified by a bioamplifier (MEB-9404, Nihon Kohden Corporation, Tokyo, Japan). One hundred responses were averaged, and the sampling rate was once 0.1 ms. The a-, b-, and d-waves were recorded with a hardwired band pass filter set at 20–200 Hz. The PhNR was recorded with the band pass filter set at 100–500 Hz. A narrow filter for 50 Hz was used to improve the signal-to-noise ratio. These methods used to record the iFMERGs were reported in detail by Matsumoto et al. and similar procedures were used [5]. One exception was the use of 25-gauge instead of 29-gauge chandelier lighting for the background illumination. The room temperature was set at 25.0 °C throughout the operation, and the intraocular irrigating solution (BSS PLUS intraocular irrigating solution 0.0184%, Alcon Surgical, Fort Worth, TX, USA) was maintained at room temperature.

The preoperative medication was 25 mg hydroxyzine and 15 mg pentazocine, which were injected intramuscularly. Anesthesia was induced by a sub-Tenon injection of an equal volume mixture of 2% xylocaine and 0.5% bupivacaine after disinfection of the conjunctival sac. Cases of combined cataract surgery, phacoemulsification and aspiration (PEA) were performed with an implantation of an intraocular lens (IOL) in the capsular bag before the PPV. The vitrectomy was performed using 3 trocars with 25-gauge closure valve vitrectomy system and the IOP was kept at 30 mmHg during the surgery. A small amount of triamcinolone acetonide (MaQaid intravitreal injection 40 mg, Wakamoto Co., Led, Tokyo, Japan) was sprayed to make the vitreous more visible, and core vitrectomy was performed. A posterior vitreous detachment was created by suction with a vitrectomy probe (CONSTELLATION Vison System, Alcon, Bromma, Sweden) if needed. It had been reported that the temperature of the vitreous cavity influences the ERG responses significantly, particularly the peak time; therefore, it was important to maintain the temperature of the vitreous cavity during the procedure. We used a closed infusion to stabilize the intraocular temperature for 5 min just before the ERG recordings. After 5 min of light-adaptation and stabilizing of the intraocular conditions, the first iFMERGs were recorded as the baseline value (Phase 1). Then, the BBG was sprayed to stain the ILM. After washing the excess BBG floating in the vitreous sufficiently and 5 min of light adaptation and stabilization, the second iFMERGs were recorded (Phase 2). Then, ILM peeling was performed while observing the surgical field through a contact lens, and the third iFMERGs were recorded after 5 min of light-adaptation and stabilization (Phase 3). After that, the necessary treatment for each disease, including shaving, was performed appropriately and the operation was terminated.

iFMERGs were recorded twice at each phase. The implicit times and amplitudes of the a-wave, b-wave, PhNR, and d-wave were measured. The average of the two measurements at each phase was used for the statistical analyses. Wilcoxon signed-rank tests were used to determine the significance of the differences at Phase 2 and 3 in comparison with Phase 1. A *p* value of 0.05 was taken to be statistically significant. Surgeries were performed by multiple surgeons (SM, KS, GT) and analysis was completed without the information of the surgeons.

## 3. Results

Representative iFMERGs recorded during the three different phases are presented in Figure 1. The demographics of the patients and the measured values of each component of the iFMERGs are presented in Table 1 and plotted in Figure 2.

The average ± standard deviation (SD) of the implicit time of the a-wave was 24.9 ± 3.49 msec at Phase 1, 25.5 ± 3.96 msec at Phase 2, and 24.0 ± 3.46 msec at Phase 3. The average implicit times of the b-wave was 45.2 ± 3.71 msec at Phase 1, 44.7 ± 4.61 msec at Phase 2, and 44.4 ± 4.37 msec at Phase 3; that for the PhNR was 67.6 ± 6.48 msec at Phase 1, 67.7 ± 5.87 msec at Phase 2, and 66.0 ± 5.13 msec at Phase 3. For the d-wave, the average ± SD implicit time was 128.0 ± 4.24 msec at Phase 1, 127.4 ± 2.99 msec at Phase 2, and 127.3 ± 5.58 msec at Phase 3. None of the differences in the implicit times at Phase 1 vs. Phase 2, and at Phase 1 vs. Phase 3 were significant.

The average ± SD of the a-wave amplitude was 2.12 ± 1.28 µV at Phase 1, 1.86 ± 0.95 µV at Phase 2, and 2.03 ± 1.33 µV at Phase 3; that for the b-wave was 5.33 ± 3.05 µV at Phase 1, 4.97 ± 2.41 µV at Phase 2, and 5.21 ± 3.16 µV at Phase 3. The average ± SD of the amplitudes of the PhNR was 1.89 ± 1.18 µV at Phase 1, 1.60 ± 0.88 µV at Phase 2, and 1.67 ± 1.03 µV at Phase 3; that for the d-wave was 1.94 ± 0.96 µV at Phase 1, 1.70 ± 0.84 µV at Phase 2, and 1.88 ± 1.02 µV at Phase 3. None of the differences between Phase 1 vs. Phase 2, and Phase 1 vs. Phase 3 were significant (Figure 3). None of the eyes had a serious complication during or after the surgery.

## 4. Discussion

Our results showed that the BBG-dye-assisted ILM peeling did not alter the macular function as determined by the amplitudes and implicit times of the iFMERGs. Thus, we conclude that the spraying of BBG, membrane staining, and ILM peeling under irrigation and illumination are safe procedures. The effects of these systemically administered preoperative medications on the ERGs are not known, but they are expected to have minimal, if any, effects on full-field and focal macular ERG because of the small amounts in the retinal circulation. Furthermore, the ERGs were recorded within a short time frame of a few minutes before and after the ILM peeling during surgery on the same patient; therefore, the possibility of large fluctuations of the ERGs due to the effects of the medications are considered to be quite low. Ejstrup et al. injected ICG, BBG, or TA subretinally in vitrectomized porcine eyes and recorded mfERGs 6 weeks later [25]. They reported that the amplitudes of mfERGs were decreased, and the implicit times were prolonged only in eyes that were injected with ICG. Machida et al. recorded FMERGs before and 1, 3, 6, 9, and 12 months postoperatively in eyes that underwent PPV with dye-assisted ILM peeling for an MH [19]. They used ICG, BBG, or TA, and they reported that the amplitudes of all components gradually increased with time after surgery. The implicit times of the a- and b-waves were significantly prolonged at 1 month and then gradually returned to the baseline times. No significant differences were found in these changes among the groups. They concluded that none of the three agents were toxic to macular function. A literature review on the visual acuity after BBG-assisted ILM peeling for an MH showed that BBG could contribute to better visual acuity outcomes than other dyes after peeling the ILM in patients with an MH [26,27]. Although we did not measure or evaluate the long-term effects of BBG, these findings are consistent with our results in that BBG-assisted ILM peeling had no significant effect on macular function.

In contrast, Terasaki et al. recorded FMERGs in eyes with an MH that had undergone PPV with (ILM-off group) and without (ILM-on group) TA-assisted ILM peeling. They observed that the amplitude of the b-wave increased significantly 6 months after surgery in the ILM-on group but not in the ILM-off group [28]. Furthermore, they found that the percentage increase in the b-wave amplitudes 6 months after surgery was significantly greater in the ILM-on group that in the ILM-off group. They concluded that the selective delay in the recovery of the b-wave of the FMERGs 6 months after surgery suggested an alteration of retinal physiology in the macula region. Tari et al. reported a regional correspondence of reduced responses in the mfERGs in the ILM peeled area of eyes with an idiopathic macular pucker 3 months after PPV with ILM peeling without using dye [23]. These results suggested that ILM peeling itself independently from dyeing may influence the retinal function.

In our study, the differences between the Phase 2 and Phase 3 responses immediately before and after the ILM peeling were not significant. This should be interpreted with caution, bearing in mind that it was obtained from a small sample size. Taken together, it can be assumed that macular dysfunction is independent of the use of dye, and that it occurred gradually at some time after the procedure.

A question then arises as to when the temporal reduction in macular function occurred. It must be between ILM peeling and one month after the peeling. Further investigations with FMERG or multifocal ERG recordings shortly after the surgery would be helpful in determining the timing of the functional impairment.

Our study has several limitations. First, the sample size was small. If we examine the individual results in more detail, there were cases where the amplitude increased or decreased (Figure 2). Further studies with a larger number of cases are needed to investigate their characteristics by sub-analysis would be of value. Second, during surgery, the retina was exposed to low temperatures, changes in the composition of intravitreal environment, adaptation, and intraocular pressure, and combinations of these, which is far from the physiological state. Thus, the results of intraoperative ERGs and outpatient ERGs cannot be simply compared. Therefore, we made all recordings in the limited situation of surgery and compared the changes, before and after the dye spraying, dye staining, and ILM peeling, with the same procedure at the baseline. We also reported earlier that core vitrectomy changed the ERGs [5,6] and considered that one of the causes was the effect of temperature decrease due to the irrigating solution. Vitrectomy lowers the temperature of the vitreous cavity, which is enough to make the ERGs abnormal [2]. This is because the temperature of the irrigating solution is room temperature, which is lower than body temperature. We did wait for about 5 min after rinsing the sprayed BBG to minimize this effect with the assumption that the intravitreal conditions would stabilize. To ensure the same conditions, all adaptations were performed in the same room, on the same bed, with the eyes opened and the pupil dilated, and exposed to room light for 5 min. It is unknown whether stopping infusion for 5 min is enough for stabilizing the retinal temperature. Some investigators measured midvitreous temperature during vitrectomy and reported that the duration of membrane peeling was not significantly correlated with the temperature increase [29,30]. In the clinical setting during vitrectomy, it might be difficult to determine the time to stabilize the retinal temperature, and 5 min was practical. We should keep in mind the possibility of temperature effect on the current results. Third, we could not compare the iFMERGs using ICG staining as control. It is ethically not possible because ICG is retinotoxic, and BBG had already become commonplace for staining ILM at the time of this study. Fourth, no significant change does not mean non-inferiority. It is difficult to define a clinically meaningful ERG margin based on animal studies and previous reports of iFMERGs because the measurement equipment was special and there was no previous study. The current data do show no clear adverse effects, which we think is meaningful. Fifth, although the time required for membrane peeling and its area were almost the same, quantitative analysis is impossible due to its retrospective nature. Further study that collects these quantitative data is needed to remove bias from these influences.

Considering the fact that the examination time is a considerable proportion of the total surgical time, and that special equipment is required, widespread use of this test in clinical practice is difficult and not realistic. In addition, we have not found any significant changes of the surgical method according to the information obtained with iFMERG. On the other hand, the spread of intraoperative OCT has made intraoperative macular structure evaluations possible, and new macular disease treatments such as autologous retinal transplantation to the macula, macular cystectomy, and tPA injection into the central retinal artery have been developed, a system that can objectively and safely evaluate macular function intraoperatively has the potential to bring important implications for the development of surgical strategies and procedures.

## 5. Conclusions

In conclusion, BBG is safe as determined by an assessment of the macular function by iFMERGs. The iFMERGs allowed an in situ assessment of the macular function during vitrectomy and was useful for evaluating the safety of the intraoperative procedures.

## Figures and Tables

**Figure 1 jcm-11-06404-f001:**
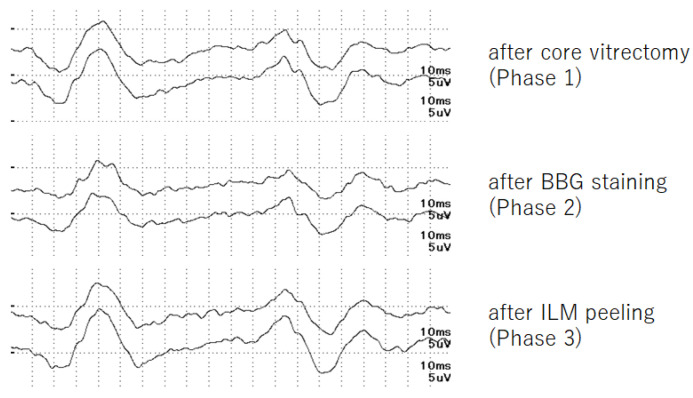
Representative iFMERG graph.

**Figure 2 jcm-11-06404-f002:**
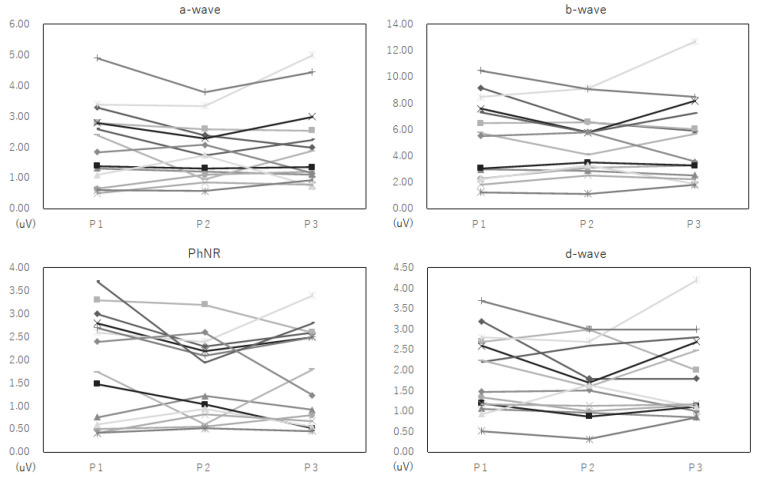
Polts of the amplitudes of each component of the iFMERGs.

**Figure 3 jcm-11-06404-f003:**
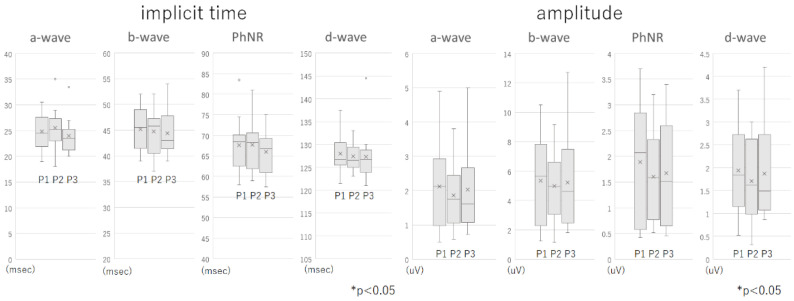
Boxplots of the implicit times and the amplitudes of the intraocular focal macular electroretinograms. No significant differences were observed between Phase 1 vs. Phase 2 or between Phase 1 vs. Phase 3.

**Table 1 jcm-11-06404-t001:** The demographic of the patients and the measured values of each component of the iFMERGs.

No.	Age	Gender	Disease	Operation	Implicit Time												Amplitude											
					a-Wave			b-Wave			PhNR			d-Wave			a-Wave			b-Wave			PhNR			d-Wave		
					P1	P2	P3	P1	P2	P3	P1	P2	P3	P1	P2	P3	P1	P2	P3	P1	P2	P3	P1	P2	P3	P1	P2	P3
1	73	Male	ERM	PPV + PEA + IOL	26.5	24.5	25	45.5	45.5	47	64.5	69	66	128.5	124	127	3.3	2.4	2	9.2	6.6	5.9	3	2.3	2.6	3.2	1.8	1.8
2	77	Male	ERM	PPV + PEA + IOL	24	26.5	23.5	45.5	43	47.5	69	67.5	72	127.5	129	126.5	2.8	2.6	2.55	6.5	6.6	6.05	3.3	3.2	2.6	2.7	3	2
3	66	Male	ERM	PPV + PEA + IOL	28	21.5	20	49	47	45	69.5	71	69	133.5	127	128.5	1.32	1.2	1.12	3	2.9	2.56	0.76	1.22	0.92	1.06	0.96	0.86
4	65	Female	ERM	PPV + PEA + IOL	27.5	25	21.5	43	46.5	42.5	68	70	69	125.5	126	123	2.8	2.3	3	7.6	5.8	8.2	2.8	2.2	2.5	2.6	1.7	2.7
5	56	Female	ERM	PPV + PEA + IOL	22	18	24	41.5	37	39	64.5	59	59	121.5	123	125	3.4	3.35	5	8.5	9.15	12.7	2.6	2.4	3.4	2.8	2.7	4.2
6	69	Male	MH	PPV	30.5	35	33.5	52	52	54	74.5	70.5	75	137.5	129	121	0.65	1.1	1.2	2.3	3.1	3.3	0.5	0.55	0.8	1.35	1	1.15
7	78	Male	ERM	PPV + PEA + IOL	26	25.5	27	45.5	46	48.5	70.5	69.5	68	129	125	130	4.9	3.8	4.45	10.5	9.1	8.5	2.7	2.1	2.5	3.7	3	3
8	62	Female	MH	PPV + PEA + IOL	24	24	23.5	39	39	42	61	62	63	126	126	125	2.6	1.75	2.25	7.35	5.8	7.25	3.7	1.95	2.8	2.2	2.6	2.8
9	57	Male	ERM	PPV + PEA + IOL	21.5	23	20.5	41.5	38.5	39.5	58	60	57.5	124	126	123.5	2.4	0.96	1.88	5.76	4.12	5.68	1.74	0.6	1.8	2.24	1.6	2.48
10	78	Male	ERM	PPV + PEA + IOL	25	26.5	20.5	41.5	41	42	61	61.5	67.5	125.5	125	124	1.84	2.08	1.16	5.52	5.84	3.56	2.4	2.6	1.24	1.48	1.52	1
11	80	Female	ERM	PPV + PEA + IOL	20	23	25	44.5	44.5	39.5	63	66.5	60.5	125.5	128.5	127	1.4	1.32	1.36	3.08	3.52	3.28	1.48	1.04	0.52	1.2	0.88	1.12
12	53	Female	ERM	PPV + PEA + IOL	24	28.5	23	49	48	49.5	83.5	81	70	129.5	133	129.5	1.1	1.75	0.73	2.25	3.3	1.88	0.6	0.95	0.55	0.93	1.64	1.1
13	77	Male	MH	PPV + PEA + IOL	30	27	26	49.5	46	43.5	70	66.5	61	133	131.5	144.5	0.5	0.86	0.78	1.84	2.54	2.24	0.42	0.82	0.68	1.18	1.14	1.18
14	73	Male	ERM	PPV + PEA + IOL	19	29	22.5	45.5	52	42.5	70	73.5	66	126	131	127.5	0.61	0.58	0.94	1.25	1.16	1.82	0.42	0.52	0.46	0.52	0.32	0.86
				Average	24.9	25.5	24	45.2	44.7	44.4	67.6	67.7	66	128	127.4	127.3	2.12	1.86	2.03	5.33	4.97	5.21	1.89	1.6	1.67	1.94	1.7	1.88
				SD	3.49	3.96	3.46	3.71	4.61	4.37	6.48	5.87	5.13	4.24	2.99	5.58	1.28	0.95	1.33	3.05	2.41	3.16	1.18	0.88	1.03	0.96	0.84	1.02

ERM: epiretinal membrane; MH: macular hole; PPV: pars plana vitrectomy; PEA: phacoemulsification and aspiration; IOL: intraocular lens implantation; SD: standard diviation.

## Data Availability

The data are not publicly available due to personal information protection.

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
