# Peer review of "Electroretinographic Assessments of Macular Function after Brilliant Blue G Staining for Inner Limiting Membrane Peeling"

_jcm, 2022, doi:10.3390/jcm11216404_

Round 1

Reviewer 1 Report

Authors present a study concerning the impact of Brilliant Blue G staining for ILM Peeling surgery. Below please find my comments.

1. The correct translation of ILM is Inner Limiting Membrane, not Internal.
2. Was the surgery performed only in local anesthesia? What was premedication? The systemic medications applied before and during surgery may affect the ERG recordings.
3. What were ERG conditions? How strong flashes etc. How the comparable light adaptation was ensured?
4. Please include representative ERG graphs in the figures.
5. Please include statistics in your results and graphs, including p values.
6. Was the flicker recording done? In fact this is the most sensitive ERG marker for immediate retinal toxicity.
7. In such short manuscript there is no need for supplementary figures, please include them in manuscript.
8. Was there any follow up ERG performed with patients? Would be nice to see the long therm comparisons of retinal function.

Reviewer 2 Report

Authors developed a new device, intraoperative focal macular electroretinograms (iFMERGs), and recorded ERG before and after Brilliant Blue G staining for ILM.   ICG is a stain still in use and could be used as a positive control. Although it is ethically problematic to dare to use ICG for research at the same facility, could we ask a facility that has been using ICG for many years to measure the ERG? It would take several months to prepare a positive control, and shouldn't they at least mention in the limitation section that there is no positive control?   Why not define a clinically meaningful ERG margin based on animal studies and previous reports of iFMERGs and perform the same statistical analysis as in non-inferiority trials to find significant differences? If it is not significant, then increase the number of cases as necessary.   I agree that iFMERGs are excellent for clinical research, but where do they fit into clinical practice? I would like to see iFMERGs reveal intraoperative retinal dysfunction and present a situation in which it can be restored by some manipulation. A normal vitrectomy takes less than 30 minutes, and the disadvantage of adding 5 minutes to that time is significant. The cost is not very low, it is not cost-effective, it is difficult to be covered by insurance reimbursement, for example, and can it be reimbursed by the same insurance reimbursement as conventional ERGs?

Reviewer 3 Report

Very well written manuscript. But the sample size is small for any meaningful conclusion. I have no further comments.

Minor comment-

Please comment whether all surgeries were performed by a single surgeon

Round 2

Reviewer 1 Report

Im satisfied with authors' responses.

Author Response

Thank you for your valuable comments.